# The Perceived Restorativeness of Outdoor Spatial Characteristics for High School Adolescents: A Case Study from China

**DOI:** 10.3390/ijerph19127156

**Published:** 2022-06-10

**Authors:** Xiaoxia Bai, Xinxin Li, Ding Yan

**Affiliations:** 1School of Architecture and Urban Planning, Huazhong University of Science and Technology, Wuhan 430074, China; baixiaoxia@hust.edu.cn (X.B.); yanding@hust.edu.cn (D.Y.); 2Hubei Engineering and Technology Research Center of Urbanization, Huazhong University of Science and Technology, Wuhan 430074, China

**Keywords:** restorative environment, high school, outdoor space, perceived restorativeness scale, spatial openness, spatial characteristics

## Abstract

Heavy schoolwork and overpopulated classrooms have made high schools overstressed environments. Previous investigations have identified a wide body of naturally restorative elements. However, evidence regarding the relationship between spatial typology and its perceived restorativeness (PR) for adolescents is limited. This paper explores the connection between spatial characteristics and PR by linking their restorative quality to how they are actually used. A high school with multiple types of outdoor spaces is used as a case study and typical spatial characteristics (area, distance, and openness) are quantified. A revised perceived restorativeness scale (RPRS) is exploited to assess the restorative quality of different spaces, and a self-reported questionnaire is used to map the actual usage. The obtained results reveal that: (1) the restorativeness of the selected spaces varied considerably, with a natural garden being more restorative than a built environment; (2) the area and openness were positively correlated to the PR, but the distance was negatively correlated; (3) the theoretical dimensions of “getting away” at high school are primarily psychological, not physical; (4) the actual use of outdoor spaces during breaks does not match the students’ favorite places or their PR. These findings expand our understanding of the role of spatial characteristics in PR in high schools and indicate direct links between campus design and restorative quality.

## 1. Introduction

Mental health problems and the learning efficiency of high school students pose ongoing challenges. Heavy schoolwork, fierce learning competition, a high concentration of students, and excessive pressure to conform have turned high schools into overstressed environments. This is especially the case in high-density urban schools, leading, in turn, to destructive effects on student health and learning. A restorative break environment contributes significantly to both physical and psychological health. It can preserve clever minds, maintain active capability, promote social interaction, and increase school-based learning efficiency [1,2]. All environments can have a certain restorative effect, but this varies [3]. Existing investigations suggest that the restorative effect of natural environments is better than built environments [4]. This has led many studies to promote the use of natural restorative elements, such as green spaces (e.g., trees, grass, forests, and parks) [5], blue spaces (featuring water) [6], and other natural elements (e.g., lighting, and visible sky) [7,8]. However, purely natural environments are often inaccessible during ordinary school breaks, and urban schoolyards are typically characterized by hard surfaces with little or no greenery [9]. Built environments form a critical part of school settings. So, despite the value of natural elements, other factors cannot be set aside.

Research has found that short-term stress has a very different effect compared to stress accumulated over the longer term, so laboratory-based experiments [10] are not suitable for the investigation of accumulated stress, especially amongst adolescents. Additionally, the restorative benefits of short-term exposure to fresh environments are different to those of long-term repeated exposure [11]. Restorative spaces therefore feature within two distinct categories: familiar daily environments for recurrent experiences (such as schools, residential spaces, and workplaces) [11,12,13] and short-term environments for intermittent experiences (such as parks, streets, squares, hotels, and hospitals) [14,15,16]. The former category relates to specific and relatively fixed user groups, while the latter is open to broader user experience. Most of the extant studies have been conducted on adults [17], such as college students, ordinary citizens, workers, and tourists. Far fewer studies relate to minors, making the relevance of current academic research to restorativeness design in schools very limited. This is a serious omission because outdoor school environments are subject to long-term and high-frequency exploitation.

Spatial typologies offer direct guidance to designers for the regulation and improvement of environmental experiences. Research has already revealed spatial factors that have a relationship with restorative environments. These include street characteristics, the scale of squares, and the degree of enclosure. A number of spatial prototypes for schools have been documented, but they have not attracted much research attention, lack consideration of restorativeness, and do not provide design guidelines. Going outside is fundamental to the perception of outdoor spaces and for their restorative effects to have any impact, and research has suggested that the quality of restorative environments can affect the occurrence of outdoor physical activities at school [18,19] and students’ cognitive performance [20]. There is thus a mutual link between environmental preferences and their restorative quality. Previous research has focused on the restorative quality of the environment, but its actual use is considered independently. According to the theory of environmental behavior, behaviors are a direct result of environmental impact. The spatial distribution of behavior may therefore be affected by a range of spatial factors, including location, scale, and quality.

This paper draws upon Kaplan’s restorative environment theory, which puts forward four dimensions of environmental restorativeness: getting away; fascination; extent; and compatibility [3,21]. The relationship between spatial typologies and perceived restorativeness will be explored across these four dimensions, which will help to make their mutual relevance clear. The ultimate goal is to consider how spatial typologies could be coupled with restorativeness when building high school environments to maximize their restorative quality and promote activities outdoors.

## 2. Materials and Methods

### 2.1. Overall Process

As this is an exploratory study, no specific hypotheses were formulated. Instead, we undertook a four-step process. In the first step, spatial prototypes were devised based upon existing mappings in the literature. Then, a large case with multiple spatial prototypes was selected and its spatial characteristics were quantified according to construction drawings provided by the school administration. In the third step, the restorativeness of the selected spatial scenes was measured via a revised perceived restorativeness scale (RPRS). Finally, the actual use of various spatial types during school breaks was methodically collected. Potential correlations were then established on the basis of the acquired results.

### 2.2. Step 1: Outdoor Spatial Prototypes of High Schools

Herein, the ‘outdoor environment’ refers to the area bounded by school buildings and enclosing campus walls. Our first step was to extract spatial prototypes of schools. To do this, we used a typology that enabled us to categorize multiple cases [22]. Through a combination of Baidu mapping data and all published instances of school buildings in the Architectural Journal from 1954 to 2020, which is the most authoritative journal in Chinese architecture, 54 relevant cases were identified. Aspects of the Design Code for Primary and Secondary School Buildings in China were also drawn upon, e.g., “It is suggested that fixed classrooms in primary and secondary school buildings shall not be set above four floors”, “The distance between the outer window of classrooms and the edge of the playground shall not be less than 25 m ”, “The sunshine entering full windows in fixed classrooms on the winter solstice shall not be less than 2 h”, and “It is suggested that the playground perimeter of a high school should be 400 m”. This background material gave us the bounding limits for the prototypes and the scale of the courtyards. Overall examination of the relevant cases revealed that a school’s outdoor spaces can be summarized as: enclosed courtyards; semi-enclosed courtyards; and open spaces, such as playgrounds, paved squares, gardens, roof platforms, and exterior corridors.

### 2.3. Step 2: Quantification of the Spatial Characteristics

To explore the relationship between the spatial prototypes and their restorative quality, it was essential to be able to quantify their spatial features. A detailed literature survey revealed that scale, location, and spatial patterns were the most accessible measures for describing outdoor spaces [12,23,24].

#### 2.3.1. Spatial Scale

As the chosen scenes did not give vertical dimensions and all the buildings had four or five floors, the spatial scale was quantified and represented by a plane area. This took the typical boundary of the space as the basis for calculation, i.e., the boundary of the building, the courtyard wall of the campus, the boundary of the platform, and the domain divisions for different teaching units.

#### 2.3.2. Distance and Spatial Location

Distance was exploited to describe the relative location between an outdoor space and the classrooms. This is a key concern in the early stages of campus design. Distance also impacts the accessibility of outdoor spaces and the extent to which they are used. This study quantified the distance between teaching buildings and the center of the outdoor space to describe position.

#### 2.3.3. Openness and Spatial Patterns

Visual outdoor spatial patterns are three-dimensional concepts that relate to the plan shape, the width to height ratio (D/H), the ratio between the enclosing entity length and the overall side length (E/T), the top shelter to total area ratio (S/A), and spatial configuration. For each individual indicator, the pattern was easy to quantify. However, one single indicator is limited when describing the overall sense of space. For instance, although Yoshinobu Ashihara’s D/H index [25] for describing outdoor space is widely used, it simplifies the real visual experience, and gives a static perspective that does not reflect users’ overall spatial perception. In contrast, spatial openness is a comprehensive index that defines a place’s degree of openness. Here, this just refers to assessments of physical openness based on visual perception [25,26]. Zhang has provided a dynamic method for calculating openness by simulating the range of the human visual field. In this study, the spatial openness of the landscape is quantified using four indexes: the porosity of the field of view; the distance of occluding objects; the height of occluding objects; and the proportion of vision. Taking a 24 × 24 × 6 m cube-like space as an example (see Figure 1), the observer’s position is at the center of the cube, a height of 1.5m is taken to form the horizon, and the surrounding view is divided into four 90° angles. To assess the visible surroundings, the openness in each direction is calculated separately (*O_h_*) and the average value is taken as the openness when looking around (*O_a_*). As there is never any construction in the upward view, the upward degree of openness (*O_u_*) is taken to be 100%. Ultimately, the spatial openness is equal to the average of *O_a_* and *O_u_*. For the above example, this can be calculated as follows:*O_h_* = [*S_s_* + *S_g_* + *S_o_* × *K_d_*(1 + *K_h_*)]/*S_v_* × 100% = [2.428 + 8.569 + 6∗12/110)]/16.997 = 68.6%(1)
*O_a_* = (*O*_*h*1_ + *O*_*h*2_ + *O*_*h*3_ + *O*_*h*4_)/4 = 68.6%(2)
*O* = (*O_a_* + *O_u_*)/2 = (68.6% + 100%)/2 = 84%(3)
where *O_h_* denotes the openness in the horizontal direction; and *O_a_* denotes the average openness value for all directions. *S_s_* is the area of sky within the field of view, *S_g_* is the area of the ground within the field of view, *S_o_* is the occluded area, *K_d_* is the openness distance coefficient, and *K_h_* represents the openness height coefficient.

#### 2.3.4. The Specific Case and Spatial Scene Selection

This study concentrates on everyday experiences of space. Only students familiar with the sampled stimuli could therefore act as participants. To control for deviation between diverse participant groups, participant selection took place prior to selecting the multiple spatial prototypes. After comparison within our pre-collected samples, a large high school located in Shaanxi Province in China was chosen. This had all seven types of outdoor environments. When considering corridors and courtyards enclosed by teaching buildings, it can be challenging to separate them. We therefore treated this whole complex as one overall scene. Note that observational results from the school staff revealed that few students took breaks on the school’s paved square. This was mainly attributed to this zone being relatively independent of the teaching area and mostly used by teachers and administrators. As its use did not relate to environmental restorativeness for students, the paved square was omitted from the investigation.

This study mainly focuses on the spatial typology of the synthetic environment and green areas covered by the selected scenes. This zone formed approximately 10% of the overall area or less, apart from the natural garden scene, which was employed for the purposes of an evaluative comparison. All the scenes used by the students were selected, located, their plan form extracted, and their spatial characteristics quantified, as illustrated in Figure 2.

### 2.4. Step 3: Measuring Perceived Restorativeness

The perceived restorativeness scale (PRS), first developed by Hartig et al. in 1996 and modified in 1997, is based on Attention Restoration Theory [27]. It has been adopted in many countries and is extensively employed to measure environmental restorative qualities. Short versions of PRS incorporating 22 items, 16 items and 8 items have all been tested to confirm their viability [27,28]. Here, we established a revised short version of PRS (RPRS) that is better suited to school students and able to keep them engaged. It removed unrelated elements and merged similar ones, leaving 16 items overall that equally cover the four theoretical dimensions mentioned above: getting away; fascination; extent; and compatibility (see Table 1). “Getting away” refers to an escape from study routines, the learning environment, and pressure. “Fascination” refers to the attractiveness of an outdoor space. “Extent” is concerned with how much space is offered, without requiring too much mental and physical strength to perceive it. “Compatibility” refers to the degree to which an outdoor space can support students’ intended activities and psychological needs.

All the students were familiar with the selected scenes and the RPRS was completed on the basis of their everyday experience. Photographs and brief text presentations did not therefore have to be included in the RPRS as prompts. We give these details here, however, because they make the location and fundamental spatial characteristics of the selected scenes clear for readers. The restorativeness of each scene was independently evaluated using 16 statements by means of a five-point Likert-type scale ranging from “completely disagree” to “completely agree”.

### 2.5. Step 4: Mapping Self-Reported Behavior during Breaks

Behavior mapping directly maps certain behaviors to the environment, in terms of both time and space [29]. This can help to uncover the relationship between outdoor design and the distribution of different kinds of behavior. Traditional behavior mapping focuses on results, without explaining corresponding reasons. Here, we used mapping questionnaires for the students to report their break behaviors, which were then translated into behavior maps. A campus map was given to each student to mark the most frequent places for 10 min breaks, 30 min breaks, and their favorite place. Some basic information that did not reveal the identity of the participants was then collected, including gender, age, class, physical activities during breaks, and, where relevant, why they did not go outside for breaks.

### 2.6. Participants

The subjective expressions of adolescents are often brought into question in terms of their maturity of understanding and ability to adequately express themselves. Some form of expression bias is often assumed in children. High school students typically fall between the ages of 15 and 18. They have received a good education and have good reading comprehension. After preparing the photo-based RPRS, self-reporting questionnaires were also developed, and three boys and three girls were invited to evaluate: the readability of the scale; whether the scene could be easily recognized; and whether the meaning of each item could be understood. The students were instructed to complete the RPRS according to their daily experience. The six students were able to identify the pictures and understand the questionnaire. This helped to validate the PRRS, its content and the suitability of the participants. On the basis of this, we selected a larger group of high school students.

As our interest was in the restorativeness of repeated long-term experiences of the environment, there needed to be consistency across the participants and their access to the selected scenes. All the participants therefore came from 12 classes located on the second and third floors of teaching unit 1, marked with a red dashed box in Figure 2. This made a potential cohort of approximately 500 students.

### 2.7. Data Collection and Analysis

The basic data consisted of three parts: the spatial parameters (calculated from the construction drawings); the restorativeness evaluated using the RPRS; and the self-reported break behaviors. Five hundred copies of both a colored RPRS and a mapping questionnaire were prepared by the researchers and all the materials were distributed by the school staff during October 2020 throughout teaching unit 1, with the exception of the first and fourth floors. The students had two weeks to reflect on their experiences and complete the RPRS in their free time. Overall, 287 scales were retrieved, 224 of which were viable (123 female, 101 male), giving a successful return rate of 78%. A total of 390 questionnaires were collected, 354 of which were viable (189 female, 165 male), giving a successful return rate of 91%. The gender ratio was more or less in balance and most participants were between 15 and 17 years old (97%). We also made use of some break time behavior data that were collected in another investigation and made available to us by the school administration. All the quantitative data were analyzed using SPSS 24.

## 3. Results

### 3.1. RPRS Reliability and Validity Analysis

The RPRS results are summarized in Table 1. Their reliability and validity were assessed using the Kaiser–Meyer–Olkin sample suitability test (KMO, 0.910 > 0.5) and the Bartlett spherical test (*p* < 0.001). This confirmed that the collected data fitted the conditions for factor analysis. Principal component analysis (PCA) was implemented to extract the common factors and test whether there was any match between the related items and the restorative dimensions. Common Factor (CF) 1 had a large load capacity (≥0.5) for items 1 to 4, which focus on whether the space can assist in moving from a learning state to a relaxed one. Items 1 to 4 therefore represent the theoretical dimension of “getting away”. CF 2 had a high load capacity for items 5 to 9 (≥0.5). These concentrate on how attractive the place is, so relate to the theoretical dimension “fascination”. Note that, although CF 2 had the largest load factor for items 5 and 6, CF 1 also contributed to it. Therefore, these two items are partially interrelated. Item 9, entitled “I hope to spend more time appreciating the surroundings”, originally aimed to explore the extension of space and the connection between spaces. However, the load factor reveals that students may have understood this to mean that the space was more attractive. CF 3 had a large load capacity for items 10 to 12. These were associated with whether the space could offer a rich sense of environment, relating to the theoretical dimension “Extent”. For item 12, the contribution of CF 1 was also significant. CF 4 was supposed to reflect the theoretical dimension of “compatibility”, which largely concentrates on whether the space could support the students’ physical and psychological requirements. Items 13 and 14 achieved a very good response. However, item 15 failed to reach a significant load capacity (<0.5) for any factor, though its load capacities for factors 1, 2, and 4 came close. Item 16 obtained similar load capacities for CFs 2 and 4, suggesting these, too, are interrelated. The common factor variance (CFV) for all the given items exceeded 0.4, and the cumulative variance contribution rate was about 59.76%. The Cronbach consistency reliability coefficient within the RPRS was 0.861. The correlations between each item and the total values were also analyzed and significant correlations were obtained at 0.01 for all items. To sum up, the RPRS was consistent with restorative environment theory and the selected items covered each of its key dimensions.
ijerph-19-07156-t001_Table 1Table 1RPRS items and factor analysis results.TDRPRS ItemsrFactor Load CapacityCFVCF 1CF 2CF 3CF 4GA1. I can temporarily put aside everyday routines**0.678 ******0.86**0.160.06−0.030.7622. It can relieve the stress and anxiety of my study life **0.698 ******0.85**0.160.110.020.7583. I can temporarily forget unpleasant things here**0.702 ******0.70**0.230.230.090.6054. I feel relaxed when I am here**0.707 ******0.59**0.330.290.050.542F5. There is something that attracts me**0.674 ****0.42**0.56**0.160.020.5216. I want to stay here longer**0.720 ****0.54**0.57**0.08−0.020.6207. This place is charming and attracts me a lot**0.705 ****0.33**0.65**0.250.010.5938. I may have some unexpected discoveries when I stay here**0.603 ****0.10**0.77**0.160.090.630E9. I hope to spend more time appreciating the surroundings**0.542 ****0.24**0.62**0.04−0.20.48010. I can get along well with other students here**0.436 ****0.100.02**0.82**0.010.68111. I can see, listen, and reflect on a lot here**0.531 ****0.180.21**0.71**−0.110.58512. Here I can do what I like**0.658 ****0.420.29**0.50**−0.010.510C13. I feel far away from what others expect of me**0.440 ****0.230.33−0.03**0.72**0.66914. There is nothing worth seeing here**−0.070 ****−0.1−0.28−0.04**0.79**0.71215. I feel I can integrate into this environment**0.639 ****0.430.340.040.350.42916. I will not feel alone here**0.565 ****0.1**0.50**0.070.450.464Note: TD—theoretical dimensions; GA—getting away; F—fascination; E—extent; C—compatibility; CF—common Factor; CFV—common factor variance; r: Pearson correlation coefficient between each item score and total PRS; **: correlation was significant at 0.01 (two-tailed).

### 3.2. Differences in the Restorativeness of the Selected Spatial Scenes

The restorativeness of each prototype spatial scene was evaluated independently according to the RPRS results. The means of the total restorative values for each scene and each dimension are given together with their standard deviations (SD) in Table 2. The SDs suggest that there are significant differences between the seven scenes. Further post hoc comparison of the total RPRS values was undertaken and it was found that the differences between the groups was generally significant at 0.05, except for four (see Table 3). The results revealed that the natural garden had the highest perceived restorativeness, confirming existing research [2,12]. Taking their restorative strength in descending order, we obtain: natural garden; open site; semi-enclosed courtyard; and enclosed courtyard.

### 3.3. Relationship between Spatial Characteristics and Restorativeness

The above results were used to explore how the students perceived the restorative effects of three key spatial characteristics. This covered not only correlations between the spatial characteristics and the total RPRS scores, but also how they related to specific theoretical dimensions. This was accomplished using Pearson correlation analysis. The results are shown in Table 4.

#### 3.3.1. Area

The areas of the selected scenes ranged from 243 m^2^ to 32,937 m^2^, with significant differences in gradient, spanning the range of outdoor spaces in the high school. The results suggest a positive correlation between area and total perceived restorativeness (r = 0.108, *p* < 0.01). Analysis relating to the theoretical dimensions revealed that area is primarily correlated with “getting away” (r = 0.173, *p* < 0.01), with no significant correlations for the other three dimensions.

#### 3.3.2. Distance

Distance describes the spatial relationship between the fixed classrooms and the outdoor environment. The results revealed a negative correlation between distance and total restorativeness (r = −0.092, *p* < 0.01). It was found to be related to all four theoretical dimensions. With an increase in distance, feelings relating to “getting away”, “fascination”, and “extent” reduced considerably, while a sense of “compatibility” rose. The negative correlation between physical distance and “getting away” can be explained by the fact that “getting away” has primarily psychological connotations, not physical ones.

#### 3.3.3. Openness

Openness is an inclusive index that is calculated on the basis of both spatial parameters and visual perception. The results revealed a positive correlation between openness and total restorativeness (r = 0.319, *p* < 0.01). This generated the highest Pearson correlation coefficient out of the three parameters studied. Openness was strongly correlated with “getting away” (r = 0.405, *p* < 0.01), “fascination” (r = 0.256, *p* < 0.01), and “extent” (r = 0.118, *p* < 0.01) at the 0.01 level. The correlation was weaker, however, for “compatibility” (r = 0.024, *p* > 0.05). It can be concluded that the more open an outdoor space, the more likely it will be perceived as restorative.

### 3.4. Relationship between Actual Outdoor Use and Perceived Restorativeness

#### 3.4.1. Prior Investigation of Student Break Activities

Going outside is the most direct way for restorative effects to come into play. A census (*n* = 4043) of students’ activities during break time was undertaken by the school administration in May 2020. The achieved responses show that 56% of students prefer to stay indoors for their daily breaks between classes. A total of 81% spend less than 2 h per day outdoors on average (including all outdoor activities). This suggests that the potential restorativeness of the campus outdoor environment is not being fully exploited. To further understand the barriers preventing students from going outdoors for their daily break, a multiple-choice questionnaire with an open-ended supplement was implemented. The most frequently selected reasons, in order, were: “Break time is too short to go outdoors” (*n* = 1694, 41.9%); “Seize the time to study” (*n* = 1432, 35.4%); “Sitting in the classroom can alleviate fatigue better than standing outdoors” (*n* = 1376, 34.0%); “Too many students in the corridors” (*n* = 1058, 26.2%); “Nothing interesting to see in the nearer outdoor spaces” (*n* = 953, 23.6%); and “It is too cold, too hot or too sunny” (*n* = 617, 15.3%). These reasons point to subjective issues regarding the accessibility of the outdoor environment, subjective choices, degree of personal comfort, degree of crowding, attractiveness of the outdoor environment, and the comfort of the physical environment, respectively. Creating outdoor spaces with higher restorative qualities may attract more students to go outside for a break. Most of the reasons given appear to be related to spatial characteristics.

#### 3.4.2. Self-Reported Break Behavior Mapping

A total of 354 viable copies of the mapping questionnaire were collected from the selected teaching unit. A total of 263 students declared two favorite places, and 91 students reported only one. The counting and superposition of image markers was completed by the researchers manually. The relevant proportions are listed in Table 5. It can be seen that there is a high degree of overlap between spaces and their proximity to the classrooms. The distribution of the students’ positions also indicates strong field boundary characteristics. They rarely cross over to other teaching units for their break.

For the 10 min short-term break, the top choice was the enclosed courtyard combined with corridors (55.1%). However, only 15.8% of the students reported it as their favorite place. In contrast, 62.1% of the students considered the playground their favorite place, yet only 9.6% of them actually used it for short-term breaks. Scenes 1 and 5 have a similar spatial prototype and obtained similar restorativeness scores, yet the actual usage varied significantly. This suggests that the actual usage during short-term breaks is influenced by other preferences rather than restorative quality, with distance potentially playing a dominant role.

For the 30 min long-term break, the situation changed significantly. The distribution of students was much more discrete. The proportion using the playground grew to 31.1%, while those using the enclosed courtyard decreased to 26.0%. Note that the natural garden achieved the highest restorativeness rating, with 41.5% of students reporting it as their favorite place. However, during longer breaks, the proportion of students using the natural garden only rose slightly to 11%. The playground and the natural garden were the top two favorite places, with the natural garden obtaining the highest restorative score, yet it remained under-utilized even for longer breaks. We need to better understand why actual use does not match restorative quality or even the choice of favorite scenes as this is not consistent with previous research [30]. Another popular scene was the platform, regardless of the length of break. This could be a result of its openness and proximity. It is also worth mentioning that very few students preferred to take any length of break in scenes 5, 6, and 8. This could be a consequence of their limited accessibility and functionality. To sum up, the long-term break distributions were more consistent with the students’ favorite places and restorative quality, but there were notable anomalies.

## 4. Discussion

The mental health issues and learning-induced cognitive load confronting adolescents raise serious problems around the world [1,2,12]. These are not problems that are going to go away. Understanding the relationship between spatial characteristics and restorative quality of the environments that they inhabit is essential because it will help designers to better support their psychological health and capacity to learn in the future. The improved design of daily environments such as campuses, which are vital spatial conduits of learning, may help to alleviate these problems and promote perceived restorativeness. This is the responsibility of designers, administrators, and policymakers. However, academic studies that focus on the spatial typology of such environments are still relatively rare. Our findings show that the perceived restorativeness of outdoor spaces in schools can have a significant role to play. We have also established a firm link between spatial characteristics and actual outdoor usage. Devising spatial prototypes is also essential because they can play a determining role in the perceived comfort of microclimates [31], which our general investigation of break spaces revealed to be an important influence upon the willingness of students to go outside. Accessibility, openness, and the microclimate have been combined to describe the quality of outdoor spaces in other studies [24]. All these factors are related to spatial characteristics. There is scope for future research to combine all known factors to provide a comprehensive solution to the design of a restorative campus.

We found the area of outdoor spaces is positively related to restorativeness. However, this can be restricted in high-density urban areas. The scale of land use is a challenge for urban schools, and they often have little space for greenery [32]. Studying the area of outdoor spaces can provide a solid reference for school planning and land use indexes. From the perspective of restorative dimensions, area is related to “getting away” and physical or mental escape. This includes physically distancing oneself from sources of stress and feeling psychologically away from a previous state. Outdoor areas are usually much larger than classrooms, offering a clear contrast and supporting openness, which had the highest Pearson correlation coefficient with “getting away”.

The distance of outdoor spaces from classrooms is negatively related to both restorativeness and actual use. Interestingly, the correlations between distance and the theoretical restorative dimensions revealed a negative relationship with “getting away”. This suggests that getting away psychologically plays a particularly crucial role in high schools. The existing high school system in China has regular classes in a fixed classroom. This is quite different to rotational teaching systems. The outdoor activities of students are not independent of the restorativeness of the scene, but tend to be passively distributed in adjacent areas, especially during short-term breaks. Therefore, areas close to classrooms have an essential role to play in perceived restorativeness. It is necessary to improve accessibility to environments with high restorativeness and to seek to promote restorativeness across all areas equally. The distance effect reported here resembles previous findings on the outdoor play of children [33], where it was found that the frequency of access to play spaces decreases exponentially with distance.

The openness of outdoor spaces is positively related to restorativeness. The playground is typically the largest outdoor area in a high school. Arranging the spatial layout between the teaching area and the playground might provide a way of making full use of the openness of the playground to enhance restorativeness. The playground and the natural garden were the students’ two favorite places. However, analyzing the actual use of these two places made it clear that the playground was much more popular than the garden. This suggests that access to the natural garden is not more important than the playground, which is actually more open and compatible with the physical activities of adolescents. Prior research has argued that the volume of greenery is strongly related to the perceived restorativeness of playgrounds [34]. Both spatial and natural effects should therefore be superposed during the design process [35].

There are several theories associated with spatial characteristics and restorativeness, the most famous being Kaplan’s restorative environment theory [3]. The main advantage of this theory is that it analyzes the mechanisms that contribute to an environment’s restorativeness across four sub-theoretical dimensions, which can contribute to a deeper understanding of just what to design for. Another related theory is prospective refuge theory [36], which suggests that enclosures, such as having cover behind one’s back, is crucial for relaxation. This implies that spatial borders may affect restorative quality, so these should feature in future research. One other relevant theory is Biophilic Design, which focuses on how we perceive nature, including direct nature (such as greenery and water), indirect nature (such as naturally derived materials, colors, and shapes), and abstraction in perceiving nature (such as spaciousness, place-based relations, and inside-outside spaces) [7,37,38]. Spatial patterns are closely associated with how students can gain access to nature. The spatial characteristics (area, distance, and openness) discussed in this study have a potential connection with abstraction in perceiving nature. Biophilic theory may therefore assist in the integration of studies of restorativeness, including natural elements and fabricated ones.

With regard to the limitations of this study, in-school data collection from adolescents was a challenge. The behavioral data were collected by self-reported questionnaires. These are voluntary and one can only analyze the questionnaires students are willing to fill in, so we did not capture every student’s behavior. There are also some disadvantages with the self-reporting of behavior, such as the accuracy of the exact location. However, behavior mapping has been successfully exploited by other investigators and its effectiveness has been proven. Our purpose, here, was to directly access attributions and count proportions that could help with explaining the results. In the future, device-based technologies could be used to make the behavior mapping more accurate, such as unmanned aerial vehicles and GPS [39]. One other thing to note is that a school’s outdoor characteristics also affect the view from classroom windows. Prior research has established that window views can contribute to stress reduction and perceived restorativeness [8,40], making it more suitable for students to take a break in their classrooms. Future studies should therefore attempt to connect indoor and outdoor restorative quality. Finally, teachers in high schools also face stress, so future environmental design should consider how to provide restorative experiences that are of equal benefit to both students and teachers [41].

## 5. Conclusions

This paper has shown that outdoor spatial characteristics can influence their perceived restorativeness for adolescents in high schools. This includes both positive (area, openness) and negative (distance) influences. Therefore, not only direct natural elements, but also spatial variables should be considered when designing and evaluating campus restorativeness. Area and openness relate to spatial restorative qualities, while distance relates to accessibility, which influences how outdoor restorativeness might be brought into play. This was apparent in both the RPRS-based results and the students’ actual use of outdoor spaces during breaks.

Spatial configuration is one of the most crucial aspects of an architectural project, and the restorativeness of a campus should feature early on in its design. From a layout perspective, the playground is a necessary outdoor environment and offers a large area that can maximize openness, hence it being one of the students’ favorite places. The distance between fixed classrooms and playgrounds plays a key part in whether their restorativeness can be exploited. When planning the campus layout, the more accessible the playground, the better. From a quality perspective, the outdoor environment near fixed classrooms should be given more attention and designed with restorativeness firmly in mind. Campus restorativeness not only depends on the number of high-quality resources, but their location. When compared, it is clear that natural gardens are more restorative than built environments, so enhancing natural elements in outdoor spaces remains a good strategy. Overall, these findings should offer some solid insights for future designers and planners.

## Figures and Tables

**Figure 1 ijerph-19-07156-f001:**
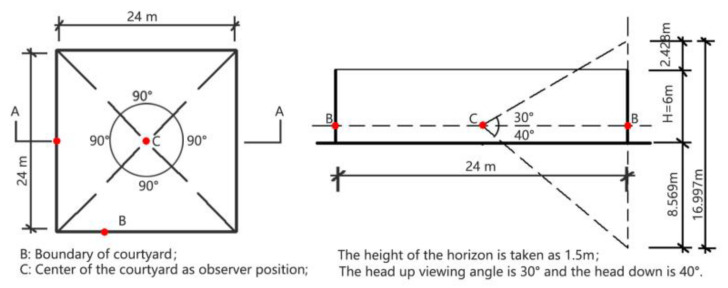
Calculating the openness factors of a hypothetical courtyard.

**Figure 2 ijerph-19-07156-f002:**
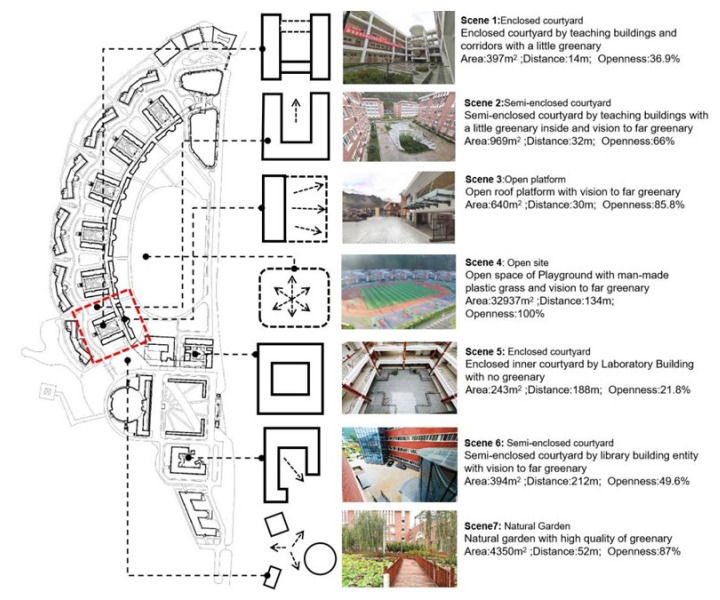
The selected scenes, their location, and their spatial characteristics.

**Table 2 ijerph-19-07156-t002:** Mean RPRS values and standard deviations for all spatial scenes.

Scene Code.	Total PRS	Restorative Feature Dimension			
		GA	F	E	C
	Mean (SD)	Mean (SD)	Mean (SD)	Mean (SD)	Mean (SD)
Scene 1	3.15(0.43)	3.06(0.62)	3.19(0.62)	3.62(0.70)	2.53(0.67)
Scene 2	3.30(0.50)	3.40(0.70)	3.38(0.72)	3.52(0.70)	2.60(0.80)
Scene 3	3.56(0.56)	3.78(0.76)	3.65(0.71)	3.73(0.75)	2.61(0.98)
Scene 4	3.51(0.60)	3.79(0.86)	3.52(0.77)	3.68(0.76)	2.70(1.02)
Scene 5	3.21(0.54)	3.09(0.68)	3.32(0.71)	3.53(0.74)	2.70(0.86)
Scene 6	3.31(0.54)	3.39(0.75)	3.36(0.68)	3.53(0.72)	2.71(0.82)
Scene 7	3.90(0.53)	4.17(0.72)	4.13(0.66)	3.90(0.67)	2.77(0.91)

Note: SD—standard deviation; GA—getting away; F—fascination; E—extent; C—compatibility.

**Table 3 ijerph-19-07156-t003:** Values and standard deviations for all spatial scenes.

Scene Code.	Scene 1	Scene 2	Scene 3	Scene 4	Scene 5	Scene 6
Scene 2	0.15 *					
Scene 3	0.40 *	0.25 *				
Scene 4	0.36 *	0.21 *	−0.04			
Scene 5	0.06	−0.09	−0.35 *	−0.30 *		
Scene 6	0.16 *	0.01	−0.24 *	−0.20 *	0.10 *	
Scene 7	0.75 *	0.60 *	0.34 *	0.39 *	0.69 *	0.58 *

Note: LSD test was conducted, where: * the differences were significant at 0.05 (two-tailed).

**Table 4 ijerph-19-07156-t004:** Correlation between spatial characteristics and perceived restorativeness.

Spatial Characteristic	Correlation	Correlation for Restorative Feature				
		Total PRS	BA	F	E	C
Area	r	0.108	0.173	0.047	0.037	0.026
	*p*-value	0.000 **	0.000 **	0.062	0.139	0.299
Distance	r	−0.092	−0.101	−0.090	−0.078	0.052
	*p*-value	0.000 **	0.000 **	0.000 **	0.002 **	0.038 *
Openness	r	0.319	0.405	0.256	0.118	0.024
	*p*-value	0.000 **	0.000 **	0.000 **	0.000 **	0.351

Note: Pearson correlation analysis was adopted: r—Pearson correlation coefficient; ** correlation significant at 0.01 level; * correlation significant at 0.05 level; GA—getting away; F—fascination; E—extent; C—compatibility.

**Table 5 ijerph-19-07156-t005:** Self-reported behavior during breaks and favorite places.

Scene Code.	Short-Term Break		Long-Term Break		Reported Favorite Places	
	Number	Rate	Number	Rate	Number	Rate
Scene 1	195	55.1%	92	26.0%	56	15.8%
Scene 2	29	8.2%	20	5.6%	37	10.5%
Scene 3	64	18.1%	74	20.9%	141	39.8%
Scene 4	34	9.6%	110	31.1%	220	62.1%
Scene 5	0	0	1	0.3%	2	0.6%
Scene 6	0	0	4	1.1%	2	0.6%
Scene 7	31	8.8%	39	11%	147	41.5%
Scene 8	1	0.3%	14	4%	12	3.39%
Mapping Results	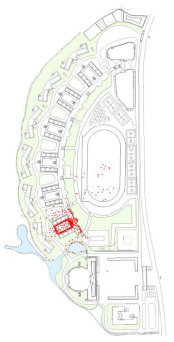		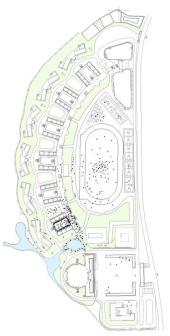		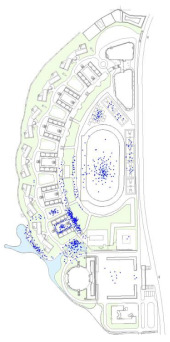	

Note: Red dots represent the places used most frequently during 10 min breaks; Black dots represent 30 min breaks; Blue dots represent favorite places. Scene 8 refers to the paved square used by the teachers and administrators.

## Data Availability

Not applicable.

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
