# Peer review of "The Perceived Restorativeness of Outdoor Spatial Characteristics for High School Adolescents: A Case Study from China"

_ijerph, 2022, doi:10.3390/ijerph19127156_

Round 1
Reviewer 1 Report
This is a topic of interest as school designs often do not pay enough attention to the quality of outdoor space which as the paper shows can have great restorative value to pupils doing concentrated indoor studies.
The English is OK but has a very heavy repetitive style and needs an English editor to polish it up a lot because as it is you have to read things several times to understand it . Paragraphs too long with too many words. It is a very tiring read.
Needs a clearer explanation of RPRS and show how it was revised. The description offered is too hazy and dullened by the language style.
Conclusions need to clearly and concisely explain what planners and designers should do to improve school outdoor space based on the work of this paper including the relevant work referred to by others.
Author Response
Response 1
The perceived restorative scale (PRS), which was developed by Hartig and his colleagues in 1996 and modified in 1997 based on Attention Restoration Theory is extensively employed in measuring environmental restorative qualities and translated into German version, Spanish version, Chinese version and so on. The validity of short versions with 22 items, 16 items and 8 items were all tested. This study modified the Chinese version of 22 items into 16 items, we cut down 6 items related to identifiability or semantically similar items, Because students are familiar with all scenes and similar items are not equal for the coverage. The changed items are:
- “I want to know more about this place. “and “I'm confused here.” were cut due to students are familiar with all scenes. (2) “This place is fascinating” was cut due to similar with “This place is charming and attracts me a lot” in Chinese context. (3)“This is a good place for me” was cut because it is too general.(4)“There is a sense of belonging here” was cut because it shows similar meaning with “ I feel I can integrate into this environment” in Chinese, and is too general. (5)“It could help diverse my attention here” shows some similar meaning with “I can temporarily put aside the everyday routines” and“I can temporarily forget unpleasant things here”,and we choose the later ones because they are more specific for the students to answer.
Response 2
We do agree with the reviewers’comments and modified the description mode of conclusion.
Response 3
It is not easy for ourselves to improve the language in a short time, so we hired an english editing and polishing organization named “Editsprings” to help us. It is mentioned in the Acknowledgments.
Reviewer 2 Report
The manuscript entitled “On the role of spatial characteristics in perceived restorative-ness of adolescents: outdoor spaces of a high school as a case study” is a work that provide understanding about environmental restorative quality since this information is helpful to architects for future design.
The paper has some merits, and an interesting topic is assessed, and this reviewer consider spatial characteristics as very useful data for promoting the psychological health. However, the content is confusing in some parts and difficult to follow for an external reader. In addition, it is necessary to complete/detail information in order to understand the study properly.
The research method applied in the article is correct. However, I came to the judgment of considering it non-acceptable in the present form and before publication there are some number of issues that need to be addressed in order to improve the general quality of the paper.
In detail:
Materials and methods
• It is indicated an experimental study with four steps is conducted, however the development of the four steps is not clear. After reading the manuscript, it is indicated that subjective expression is questioned, and mapping questionnaires are prepared. It is not expressly detailed that surveys are carried out, nor is their content exposed. How are they organized, how long do they last, what is asked? Etc. This information is relevant to better understand the development of the study.
• 2.2. Outdoor spatial prototypes of high schools. Based on what parameters are the spatial prototype of schools determined? The use of the typology approach is indicated, but it is not explained how this approach has been carried out, nor is the selection made justified. Has any statistical or parametric study been done? What are the main characteristics of the selected prototypes?
• 2.3.1 area for the spatial scale. Any more detailed description about plane area? Shape, dimensions, etc.
• 2.3.3 Openness for the spatial patterns. What is the difference between Oh and Oa? It is not detailed what Oa is.
• 2.5 Participants. Line 220. “serving approximately 500 students” (as marked in Fig.2). Where is this information marked in the figure?
• Line 224 “mapping questionnaires” What are these questionnaires like? Detail please.
• Table 3. Scene 1?
Reviewer 3 Report
The topic is very interesting and from the point of view article structure I do consider that there is nothing to report. The biggest problems that I think the article has, are related primarily to the title, which seems too generic and gives the impression that it refers to all types of schools, from the multitude of cultures found around the world when it is not so, instead it is a case study done on a single school, in an area with a unique cultural specificity. The spaces in this part of the world differ both in the development of the didactic act and in the principles of use and size, being able to be considered a unique typology, specific to a place. I consider that a change of title to emphasize this aspect is essential.
Another problematic element is that this article draws general conclusions from a single case study that refers to a single school, with a certain context, with a certain type of neighborhood, with a certain type of sunshine, the results can thus be erroneous if we think about a generalization. One last aspect that deserves to be considered is related to the architectural qualities of the analyzed outdoor spaces, they are not all the same, with the same finishes, the same features, including class proximity and preferences towards colleagues in certain classes that can lead to the preference of one place over another. It is possible that these elements are already included within the information being processed through the statistical analysis program, but still, I consider that an outdoor space can be avoided if the pavement is not good or the outdoor furniture is uncomfortable.
Author Response
Response 1:
The reviewers’ advice is accepted and the author changed the title to “The perceived restorativeness of outdoor spatial characteristics for high school adolescents: a case study from China”
Response 2:
On the one hand, case study is a particular way of observing any natural phenomenon or case, and the defect is that it can not reach a universal conclusion from a case. We do agree with the reviewers’ comments and modified the description mode of conclusion. On the other hand, This study concentrates on daily repeated experiences; therefore, only students familiar with the stimuli samplings could be selected as participants. we selected a campus with multiple spatial types, the exact advantage is students came from same region with same educational background and same age period. It could help avoid some external factors as much as possible.
Here we need to give more detailed information about this school. This is a relatively new campus built in 2015. The building interface, roads and pavement of the campus are basically intact; There were no specialized furniture in the outdoor spaces. Buildings in the campus have uniform style with similar decoration of texture. It is possible that class proximity may affect preference.
The main purpose of this study is trying to explore the relation between spacial characteristics and restorativeness. Behavior mapping is additional for reflecting actual usage, which could be help for further understanding the importance of location and distance based on this case study. Quantitative relationships cannot be established between behavior mapping and restorativeness.
Round 2
Reviewer 2 Report
The new version of the manuscript is highly improved. The authors have implemented the requested changes raised in the first round of review by integrating the information required and carrying out a general revision of the manuscript. In addition, they have replied satisfactorily to the comments. As a consequence, this reviewer consider this contribution is certainly worth being published in the journal International Journal of Environmental Research and Public Health.
.